# Model-Independent Determination of the Cosmic Growth Factor

Sophia Haude[*1], Shabnam Salehi[1], Sofía Vidal[1], Matteo Maturi[1], Matthias Bartelmann[2]

**1** Institute for Theoretical Astrophysics, ZAH, Heidelberg University, Germany
**2** Institute for Theoretical Physics, Heidelberg University, Germany
* sophiahaude@gmx.de

December 11, 2019

## Abstract

Since the discovery of the accelerated cosmic expansion, one of the most important tasks in observational cosmology is to determine the nature of the dark energy. We should build our understanding on a minimum of assumptions in order to avoid biases from assumed cosmological models. The two most important functions describing the evolution of the universe and its structures are the expansion function $E(a)$ and the linear growth factor $D_+(a)$. The expansion function has been determined in previous papers in a model-independent way using distance moduli to type-Ia supernovae and assuming only a metric theory of gravity, spatial isotropy and homogeneity. Here, we extend this analysis in three ways: (1) We extend the data sample by combining the Pantheon measurements of type-Ia supernovae with measurements of baryonic acoustic oscillations; (2) we substantially simplify and generalise our method for reconstructing the expansion function; and (3) we use the reconstructed expansion function to determine the linear growth factor of cosmic structures, equally independent of specific assumptions on an underlying cosmological model other than the usual spatial symmetries. We show that the result is quite insensitive to the initial conditions for solving the growth equation, leaving the present-day matter-density parameter $\Omega_{m0}$ as the only relevant parameter for an otherwise purely empirical and accurate determination of the growth factor.

# 1  Introduction

The expansion function of the universe and the linear growth factor of cosmic structures are the two most fundamental functions describing the evolution of the universe and its structures. They are indirectly accessible to astronomical observations, such as luminosity-distance measurements of type-Ia supernovae (SN Ia). Combining both functions allows to distinguish between different cosmological models.

The accelerated expansion rate of the Universe has been established nearly twenty years ago based on SN Ia distance measurements [1, 2]. In the framework of the cosmological standard model, this acceleration is explained by the cosmological constant or a dynamical dark-energy component currently dominating the energy content of the universe [3]. The nature of the dark energy, however, is largely unknown. So far, all attempts to derive it from fundamental theory have led to values which are way too small to explain the cosmic acceleration. Phenomenological explanations are typically based on a dark-energy equation of state, possibly varying with time. They bypass fine-tuning problems, but lack fundamental justifications. Determining the nature of the dark-energy is among the most important tasks for contemporary cosmology. The two functions, the cosmic expansion function and the linear growth factor of cosmic structures, are the most important ingredients to investigate the nature of the dark energy.

We are here proposing a method to constrain the linear growth factor of cosmic structures without reference to any specific model for the energy content of the universe. We derive the expansion function in a way similar to that proposed by [4] and [5], but substantially simplified and standardised. The only assumptions made there are that the universe is topologically simply connected, spatially homogeneous and isotropic on average, and that the expansion rate is reasonably smooth. Extending this analysis to the linear growth of cosmic structures, we only add the assumption that the linear growth of cosmic structures on the relevant scales is locally determined by Newtonian gravity. We briefly review and revise the method of [4] in Sect. 2 and apply it to the Pantheon sample of type-Ia supernovae (SN-sample hereafter) and to the Pantheon sample combined with a sample of distance measurements from baryonic acoustic oscillations (BAO, hereafter SN-BAO-sample) to obtain a purely empirical and tight constraint of the cosmic expansion function. We describe our method to calculate the linear growth factor in Sect. 3, discuss the initial conditions for solving the growth equation, and present the results obtained from the SN-sample and the SN-BAO-sample. Finally, we summarise our conclusions in Sect. 4.

## 2 Cosmic expansion

### 2.1 Method

As outlined in [4], the expansion function can be deduced from the luminosity of light sources of known intrinsic luminosity, such as calibrated SN Ia, without assuming any specific Friedmann-Lemaître model. We briefly review this method in this Section in a modified, simplified, and standardised version.

Even though gravity is commonly described by general relativity (GR), we only need to assume that space-time is described by a metric theory of gravity. We thus treat space-time as a four-dimensional, differentiable manifold with a metric tensor $g$. Assuming spatial isotropy and homogeneity, this metric has to be of the Robertson-Walker form with a scale factor $a$. In general relativity, Einstein's field equations applied to the Robertson-Walker metric turn into the Friedmann equations, and the metric further specialises to the Friedmann-Lemaître-Robertson-Walker form. Then, the cosmic expansion function $E(a)$ is given in terms of the Hubble function $H(a)$ by

$$H^2(a) = H_0^2 \left( \Omega_{r0} a^{-4} + \Omega_{m0} a^{-3} + \Omega_{DE}(a) + \Omega_K a^{-2} \right)$$
$$=: H_0^2 E^2(a) . \tag{1}$$

This defines the cosmic expansion function $E(a)$ in terms of the Hubble constant $H_0$ and the contributing energy-density parameters. These are the radiation density $\Omega_{r0}$, the matter density $\Omega_{m0}$, the density parameter $\Omega_K$ of the spatial curvature, all at the present time, and the possibly time-dependent dark-energy density parameter $\Omega_{DE}(a)$. In the standard $\Lambda$CDM cosmology, $\Omega_{DE}$ is replaced by the cosmological constant with the density parameter $\Omega_{\Lambda0}$ at the present time.

It is important in our context that we do *not* assume any specific parameterisation of the expansion function of the type (1). Rather, we merely assume that we can build upon an underlying, but unspecified metric theory of gravity with the two common symmetry assumptions of spatial isotropy and homogeneity. The metric must then be of Robertson-Walker form, and its single remaining degree of freedom must be described by some expansion function $E(a)$ whose form is *a priori* undetermined. We reconstruct $E(a)$ from data without assuming the parameterisation (1).

As an uncritical simplification, we further assume that the spatial sections of the space-time manifold are flat, following the empirical evidence that the spatial curvature of our Universe cannot be distinguished from zero within the limits of our observational uncertainties [6]. It would be rather straightforward to extend our analysis by replacing the radial comoving distance $w$ in Eq. (9) below by the comoving angular-diameter distance $f_K(w)$.

We modify the approach developed in [4, 5] and used in [7, 8] in two important ways, allowing a substantial simplification and rendering the results much more portable than before. First, we use Chebyshev polynomials of the first kind $T_n(x)$, shifted to the interval $[0, 1]$, as an orthonormal basis-function system (see Appendix A). Second, we do not expand the distance, but a scaled variant of the inverse expansion function $E(a)$ into these polynomials.

Given measurements of distance moduli $\mu_i$ and redshifts $z_i$, with $1 \leq i \leq N$, we convert the distance moduli to luminosity distances $D_{lum,i}$ via

$$D_{lum,i} = 10^{1+0.2\mu_i} \text{ pc} \tag{2}$$

and scale the redshifts $z_i$ to the variable

$$x_i := \frac{a_i - a_{min}}{1 - a_{min}} , \quad a_i = (1 + z_i)^{-1} \tag{3}$$

normalised to the interval $[0, 1]$, where $a_{\min} = (1 + z_{\max})^{-1}$ is the scale factor of the maximum redshift in the sample. We further introduce the scaled luminosity distance

$$d_i = a_{\min}^2 (1 + \delta a x_i) D_{\text{lum},i}, \quad \delta a := \frac{1 - a_{\min}}{a_{\min}}. \tag{4}$$

Since the uncertainties on the redshifts are very small compared to those of the distance, the relative uncertainty of $d_i$ is unchanged compared to that of $D_{\text{lum},i}$. We thus obtain a scaled data sample $\{x_i, d_i\}$.

The radial comoving coordinate is

$$w(x) = \int_t^{t_0} \frac{c \, dt'}{a(t')} = \int_x^1 \frac{c \, dx'}{a(x') \dot{x}'} = \frac{c}{H_0} \int_x^1 \frac{dx'}{a_{\min} \dot{x}'(1 + \delta a x')} \tag{5}$$

in terms of the normalised scaled factor $x$. We define

$$e(x) := [\dot{x}(1 + \delta a x)]^{-1} \tag{6}$$

and use

$$\dot{x} = \frac{\dot{a}}{a_{\min} \delta a} = \frac{\dot{a}}{a} \frac{a}{a_{\min} \delta a} = H_0 E(a) \frac{1 + \delta a x}{\delta a} \tag{7}$$

to write $e(x)$ as

$$e(x) = \frac{\delta a}{E(a)(1 + \delta a x)^2}. \tag{8}$$

The luminosity distance in units of the Hubble radius $c/H_0$ is

$$D_{\text{lum}}(x) = \frac{w(x)}{a(x)} = \frac{1}{a_{\min}^2 (1 + \delta a x)} \int_x^1 dx' \, e(x') \tag{9}$$

in spatially-flat geometry, using $a = a_{\min}(1 + \delta a x)$. Thus, the scaled luminosity distance $d(x)$ is

$$d(x) = \int_x^1 dx' \, e(x'), \tag{10}$$

and the scaled, inverse expansion function $e(x)$ is its negative derivative,

$$e(x) = -d'(x). \tag{11}$$

We now proceed as follows with the transformed data set $\{x_i, d_i\}$. We expand $e(x)$ into shifted Chebyshev polynomials,

$$e(x) = \sum_{j=1}^M c_j T_j^*(x). \tag{12}$$

Then, the scaled distances $d(x)$ are given by

$$d(x) = \sum_{j=1}^M c_j p_j(x), \quad p_j(x) := \int_x^1 dx' \, T_j^*(x'). \tag{13}$$

Defining the matrix $P$ by its components

$$P_{ij} := p_j(x_i), \quad 1 \le i \le N, \quad 1 \le j \le M, \tag{14}$$

the vector $\vec{c}$ of coefficients $c_j$ is determined by the data vector $\vec{d} = (d_i)$ via

$$\vec{d} = P\vec{c} \, . \tag{15}$$

With the covariance matrix $C := \langle \vec{d} \otimes \vec{d} \rangle$ of the scaled luminosity distances $\vec{d}$, the maximum-likelihood solution for $\vec{c}$ is

$$\vec{c} = \left(P^\top C^{-1} P\right)^{-1} \left(P^\top C^{-1}\right) \vec{d} \, . \tag{16}$$

The uncertainties $\Delta c_j$ of the coefficients and $\Delta E(a)$ of the expansion function are obtained from the Fisher matrix $F = P^\top C^{-1} P$ in the following way. First, we diagonalise the Fisher matrix by rotating it into its eigenframe with a rotation matrix $R$, find its eigenvalues $\sigma_i'^{-2}$ and define a vector of decorrelated coefficient uncertainties $\Delta \vec{c}'' = (\sigma_1', \ldots, \sigma_M')$. Second, we rotate this vector back into the frame of the Chebyshev polynomials and find $\Delta \vec{c} = R^\top \Delta \vec{c}''$. The uncertainties $\Delta c_i$ obtained this way are slightly larger than the Cramer-Rao bound $F_{ii}^{-1/2}$, as they are expected to be. Beginning with a large number $M$ of coefficients, only those are kept which are statistically significant, i.e. which satisfy $|c_j| \geq \Delta c_j$.

## 2.2    Cosmic expansion function from the SN-sample

We first reconstruct the expansion function using the SN-sample of type-Ia supernovae [9], covering the scale-factor range $a \in [0.3067, 1]$. We apply the algorithm described in the preceding subsection to derive the function $e(a)$ defined in Eq. (8). Using the covariance matrix provided with the data, we determine the coefficient vector $\vec{c}$ using Eq. (16) and derive its uncertainty $\Delta\vec{c}$ from the Fisher matrix as described above. We arrive at $M = 3$ significant coefficients.

We then return to $E(a)$ via Eq. (8) and determine its uncertainty from

$$\frac{\Delta E(a)}{E(a)} = \frac{\Delta e(a)}{e(a)} \, . \tag{17}$$

This results in the expansion function and its uncertainty shown in Fig. 1. The uncertainties are very small. This is due to the fact that the entire information taken from the SN-sample is compressed into three coefficients here. The best-fitting $\Lambda$CDM model with

$$E_{\Lambda\text{CDM}}(a) = \left(\Omega_{\text{m}0} a^{-3} + 1 - \Omega_{\text{m}0}\right)^{1/2} \tag{18}$$

requires $\Omega_{\text{m}0} = 0.324 \pm 0.002$. It is shown by the red curve in Fig. 1.

## 2.3    Cosmic expansion function from the SN-BAO-sample

We repeat our analysis on the SN-BAO-sample. We collected a sample of BAO measurements by searching the literature for papers that appeared in the reviewed literature between January, 2014, and December, 2018. We selected 21 papers according to the quality and the completeness of the data description and collected 89 measurements of the angular-diameter distance $D_{\text{ang}}/r_{\text{d,fid}}$ in terms of a fiducial value $r_{\text{d,fid}}$ for the drag distance, setting the physical scale of the BAOs. The drag distance is the sound horizon at the end of the baryon-drag epoch. Of these measurements, we kept 75, removing those that seemed to be either dependent on or superseded by other measurements. These measurements fall into the redshift range $[0.24, 2.4]$ and thus extend the scale-factor range of our reconstruction of the expansion function.

The drag distance $r_{\text{d,fid}}$ is unknown to us. It is determined by

$$r_{\text{d}} = \frac{1}{H_0} \int_0^{a_{\text{d}}} \frac{c_{\text{s}}(a)\text{d}a}{a^2 E(a)} \tag{19}$$

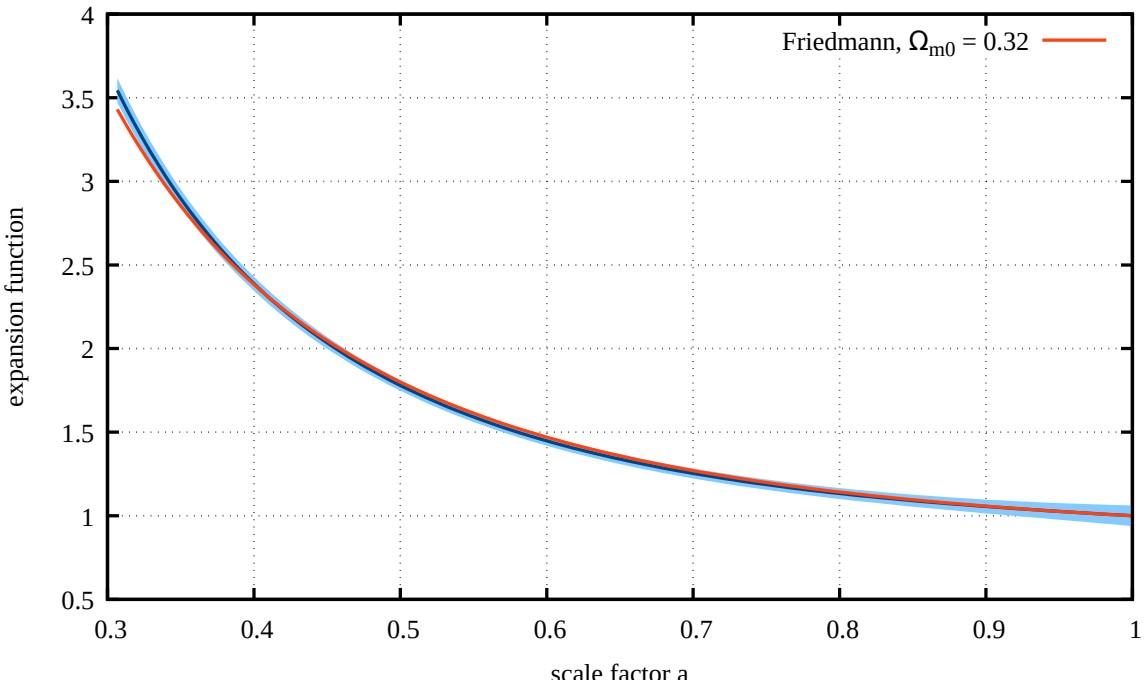

Figure 1: The cosmic expansion function $E(a)$ is shown here as reconstructed from the luminosity-distance measurements in the SN-sample. Beginning with the monomials $q_j(a) = a^{j-1}$, the model needs three significant coefficients $c_j$ whose error bars are determined by the covariance matrix of the data (see the entries in Tab. 1). The 1-$\sigma$ uncertainty shown here is so small because the entire data set is thus compressed into three numbers. The red line shows the best-fitting, spatially-flat, Friedmann expansion function.

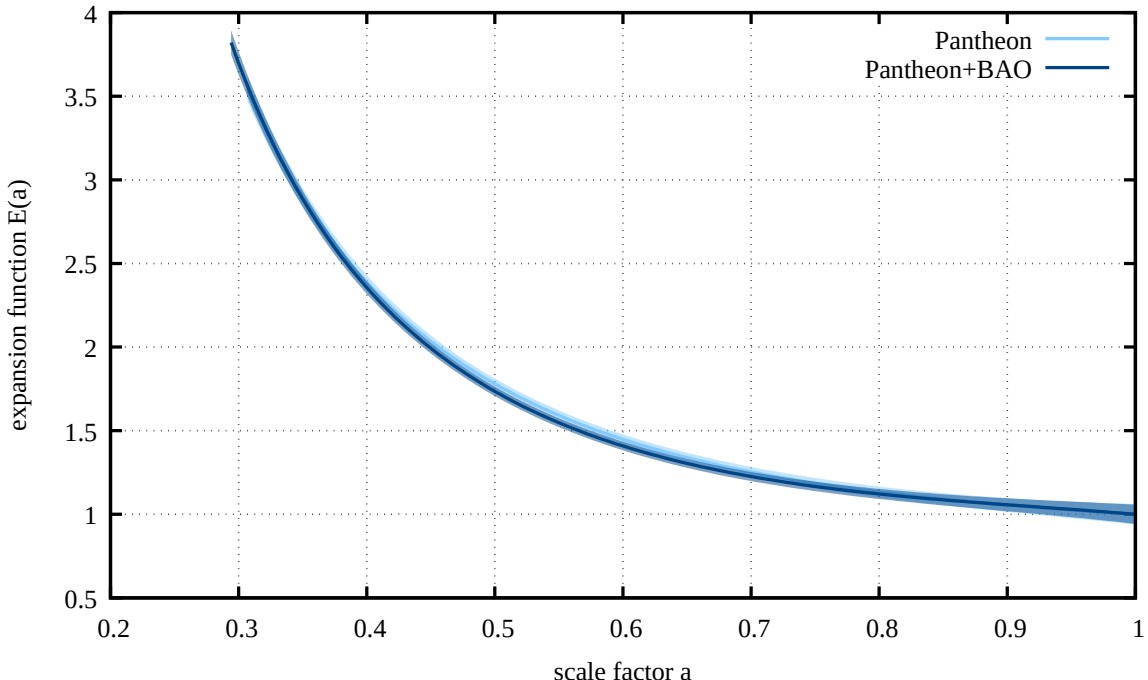

Figure 2: Expansion functions determined from the SN-BAO-sample and from the SN-sample alone for comparison. As in Fig. 1, 1-$\sigma$ uncertainties are shown. The best-fitting, spatially-flat Friedmann expansion function is the same as in Fig. 1. The reconstruction of $E(a)$ from the combined samples requires four significant coefficients (cf. Tab. 1).

and thus needs the expansion function for scale factors smaller than $a_d \approx 1100^{-1}$. In order to remain as model-independent as possible, we choose to determine $r_d$ by an empirical calibration: we applied an offset to the distance moduli corresponding to the BAO measurements such as to bring them into least-squared distance with the sample of distance moduli from the SN-sample. This offset turns out to be redshift-independent, as expected. Its value of $\Delta\mu = 10.783 \pm 0.041$ corresponds to a drag distance of

$$r_d = 143.4 \pm 2.7 \, \text{Mpc} \,, \tag{20}$$

in good agreement with the value expected in the standard $\Lambda$CDM cosmology. We further estimate the covariance matrix of the BAO data via the uncertainties quoted in the papers, combined the two statistically fully independent samples and repeated the determination of the coefficients $\vec{c}$ and the expansion function as for the SN-sample alone. The result is shown in Fig. 2. For the SN-BAO-sample, we obtain $M = 4$ significant coefficients.

Within their uncertainties, the expansion functions obtained from the SN-sample alone and from the SN-BAO-sample agree very well, but the uncertainties due to the combined sample are somewhat smaller, and the redshift range of the reconstruction is slightly extended. The fit to the standard-$\Lambda$CDM expansion function leads to a result virtually indistinguishable from the SN-sample alone, with $\Omega_{m0} = 0.319 \pm 0.002$, and is therefore not shown again in Fig. 2.

Intererestingly, the expansion function determined purely from the data is slightly more curved than the best-fitting Friedmann-Lemaître model. This difference is formally highly significant, but we do not want to emphasise it since it may be caused by systematic uncertainties in the data or their

Table 1: Significant expansion coefficients and their uncertainties

| Sample | $\vec{c}$ | | | |
|---|---|---|---|---|
| SN-sample | 0.988 | −0.372 | 0.045 | |
| | 0.033 | 0.035 | 0.018 | |
| SN-BAO-sample | 0.983 | −0.374 | 0.034 | 0.007 |
| | 0.029 | 0.032 | 0.017 | 0.001 |

interpretation. The expansion coefficients determined from both data sets, i.e. for the SN-sample and for the SN-BAO-sample, are listed in Tab. 1.

An interesting, albeit possibly premature, comparison concerns the hypothetical time evolution of the dark energy. If the expansion function $E(a)$ derived from the data were to be represented by the expansion function $E_{\Lambda\text{CDM}}(a)$ for a spatially-flat Friedmann-Lemaître model with dynamical dark energy, we should have

$$E^2(a) \stackrel{!}{=} \Omega_{\text{m0}}a^{-3} + (1 - \Omega_{\text{m0}})\, q(a) \,, \tag{21}$$

which would imply

$$q(a) = \frac{E^2(a) - \Omega_{\text{m0}}a^{-3}}{1 - \Omega_{\text{m0}}} \,, \quad \Delta q(a) = \left| \frac{2E(a)}{1 - \Omega_{\text{m0}}} \right| \Delta E(a) \tag{22}$$

for the function $q(a)$ quantifying the time evolution of the dark energy and its uncertainty. This function is shown in Fig. 3 for the SN-BAO-sample, setting $\Omega_{\text{m0}} = 0.32$ as obtained from the best-fitting, $\Lambda$CDM model determined above. It illustrates one of the advantages of our approach, as the empirically determined expansion function does not assume any specific cosmological model in general, nor a specific model for dynamical dark energy in particular.

## 3 Linear growth of cosmic structures

### 3.1 Equation to be solved

Relative to the background expanding as described by $E(a)$, structures grow under the influence of the additional gravitational field of density fluctuations $\delta\rho(\vec{x}, t) = \bar{\rho}(t)\delta(\vec{x}, t)$, where $\bar{\rho}(t)$ is the mean matter density and $\delta$ the density contrast. Structures small compared to the curvature radius of the spatial sections of the universe with a density contrast $\delta \lesssim 1$ can be treated as linear perturbations of a cosmic fluid in the framework of Newtonian gravity.

Linearising the corresponding Euler-Poisson system of equations in the perturbations and expressing spatial positions in comoving coordinates leads to the well-known second-order, linear differential equation

$$\ddot{\delta} + 2H\dot{\delta} = 4\pi G\bar{\rho}\delta \tag{23}$$

for the density contrast $\delta$ of pressure-less dust. Since Eq. (23) is homogeneous in $\delta$, the solutions for $\delta$ can be separated into a time dependent function $D(t)$ and a spatially dependent function $f(\vec{x})$, writing $\delta(\vec{x}, t) = D(t)f(\vec{x})$, where $D(t)$ alone now has to satisfy Eq. (23). Of the two linearly independent solutions of Eq. (23), one decreases with time and is thus irrelevant for our purposes. We focus on the

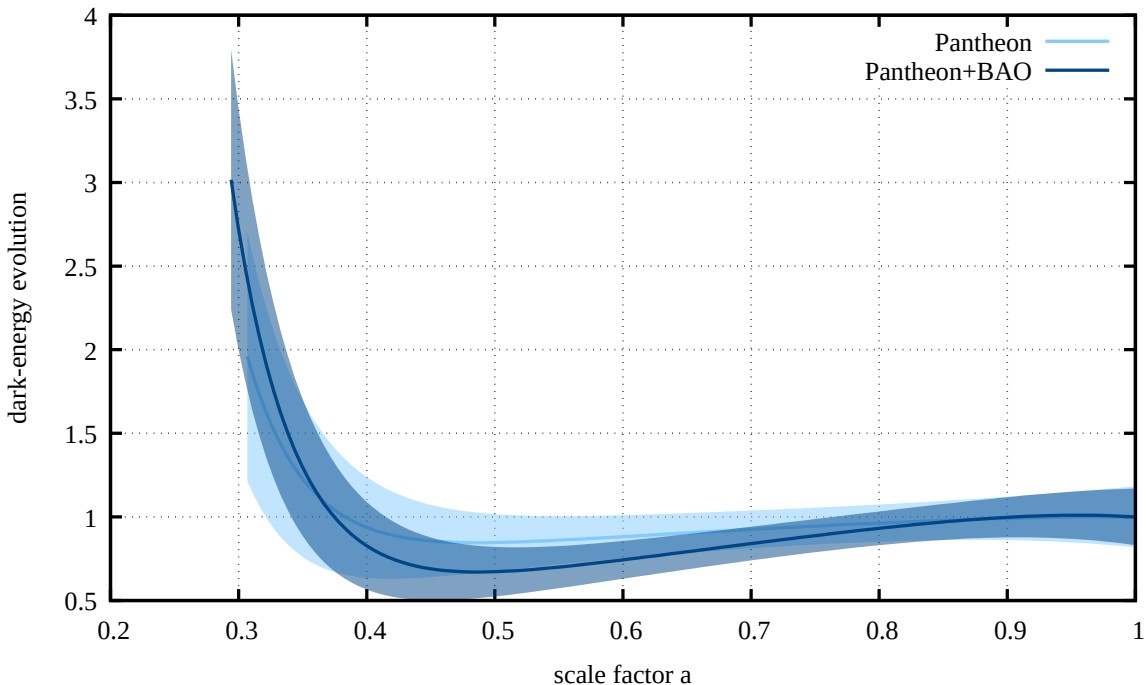

Figure 3: Constraints on a dynamical evolution of dark energy, obtained by comparing the expansion functions derived from the SN-BAO-sample with the expectation for a spatially-flat Friedmann-Lemaître model (dark blue). The light blue band shows analogous constraints obtained from the SN-sample only. As in Figs. 1 and 2, 1-$\sigma$ uncertainties are shown.

growing solution $D_+(t)$, i.e. the linear growth factor. Transforming the independent variable in Eq. (23) from the time $t$ to the scale factor $a$ then gives the equation

$$D_+'' + \left( \frac{3}{a} + \frac{E'(a)}{E(a)} \right) D_+' = \frac{3}{2} \frac{\Omega_{\rm m}}{a^2} D_+ \tag{24}$$

for the linear growth factor, where primes denote derivatives with respect to $a$.

This equation depends only on the expansion function $E(a)$, its first and second derivatives, and the matter-density parameter $\Omega_{\rm m}$. We know $E(a)$ empirically in a model-independent way from the procedure described in Sect. 2 applied to the luminosity distances of the type-Ia supernovae contained in the SN-sample, and to the distances from the SN-BAO-sample. The time-dependent matter-density parameter $\Omega_{\rm m}(a)$ is given by

$$\Omega_{\rm m}(a) = \frac{\Omega_{\rm m0}}{E^2(a)a^3} \tag{25}$$

in terms of the expansion function $E(a)$ and the present-day matter-density parameter $\Omega_{\rm m0}$.

## 3.2  Initial conditions and results for the linear growth factor

Before we can proceed to solve Eq. (24) for the growth factor, we need to set $\Omega_{\rm m0}$ and to specify initial conditions. Since we know $E(a)$ from data taken in the scale-factor interval $[a_{\rm min}, 1]$, we need to set the initial conditions at $a_{\rm min}$. Since Eq. (24) is homogeneous, the initial value of $D_+$ is irrelevant and can be set to any arbitrary value. We choose $D_+(a_{\rm min}) = 1$. Concerning the derivative $D_+'(a)$ at $a = a_{\rm min}$, we begin with the *ansatz* $D_+ = a^n$ near $a = a_{\rm min}$, assume that $n$ changes only slowly with $a$ and use Eq. (24) to find

$$n = \frac{1}{4} \left[ -1 - \varepsilon + \sqrt{(1 + \varepsilon)^2 + 24(1 - \omega)} \right] , \tag{26}$$

for the growing solution, using the definitions

$$\varepsilon := 3 + 2 \frac{\mathrm{d} \ln E}{\mathrm{d} \ln a} \quad \text{and} \quad \omega := 1 - \Omega_{\rm m}(a) . \tag{27}$$

In the matter-dominated phase, both $\varepsilon$ and $\omega$ are small compared to unity, and $n$ is approximated by

$$n \approx 1 - \frac{\varepsilon + 3\omega}{5} . \tag{28}$$

With the reconstructed expansion rate $E(a)$, the parameter $\varepsilon$ is fixed. For any choice of $\Omega_{\rm m0}$, also $\omega$ is set via Eq. (25), thus so is the growth exponent $n$, and we can start integrating the growth function with the remaining initial condition

$$D_+'(a_{\rm min}) = na_{\rm min}^{n-1} = \frac{nD_+(a_{\rm min})}{a_{\rm min}} = \frac{n}{a_{\rm min}} . \tag{29}$$

For each choice $\Omega_{\rm m0}$, we can now solve Eq. (24) with the initial conditions Eq. (29) and $D_+(a_{\rm min}) = 1$. After doing so, we normalise the growth factor such that it is unity today, $D_+(a = 1) = 1$. The uncertainty of the expansion function $E(a)$ propagates to $D_+(a)$, but the uncertainty on $D_+$ shrinks towards $a = 1$ because of the normalisation. The result is shown in Fig. 4 for $\Omega_{\rm m0} = 0.3 \pm 0.02$. The uncertainty in the growth exponent $n$ disappears in the line width of the plot.

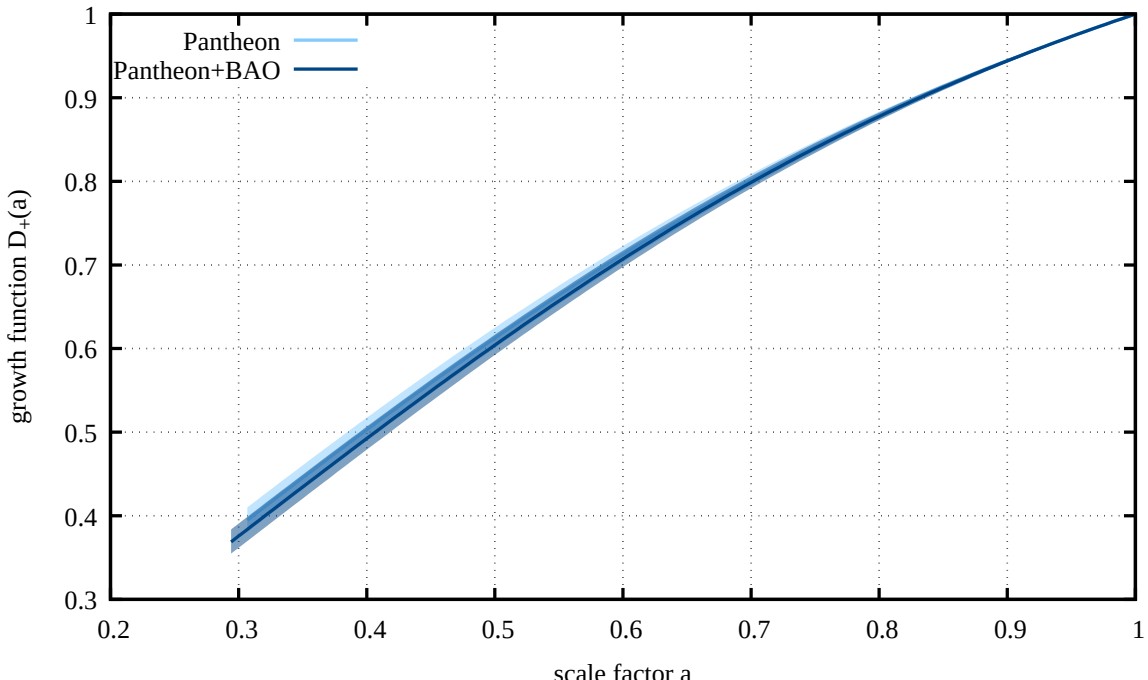

Figure 4: Linear growth factors $D_+(a)$ implied by the two expansion functions $E(a)$ shown in Fig. 2, obtained from the SN-BAO-sample (dark blue) and from the SN-sample alone (light blue). As described in the text, the growth factors are obtained by solving Eq. (25) with $\Omega_{m0} = 0.3$ based on the empirically derived expansion functions. The shaded areas cover the 1-$\sigma$ uncertainty implied by the uncertainty of the expansion function $E(a)$. Compared to the uncertainty due to $E(a)$, the uncertainty due to varying $\gamma(a_{min})$ within $[0.4, 0.8]$ is very small.

## 3.3 The growth index of linear perturbations

A common representation of the derivative of the growth factor with respect to the scale factor is given by the growth index $\gamma$, defined by

$$\frac{\mathrm{d}\ln D_+}{\mathrm{d}\ln a} =: f(\Omega_\mathrm{m}) = \Omega_\mathrm{m}^\gamma(a) \ . \tag{30}$$

Theoretically predicted values of $\gamma$ that can be found in the literature [10–17] range from approximately $\gamma = 0.4$ (for some $f(R)$ modifications of gravity [18]) to $\gamma = 0.7$. This range includes models with varying $w$ [10,16], curved-space models [15] and models beyond general relativity [10,11,18,19]. Even for models with strongly varying $\gamma$, the values for redshifts $z \in [0, 2]$ are usually very close to $\gamma \sim 0.6$.

Without further specification, Eq. (30) is obviously valid for any cosmology since the growth index $\gamma(a)$ could be any function of $a$. The substantial advantage of writing the logarithmic slope of the growth function in this way is that $\gamma(a)$ is very well constrained for a wide range of cosmological models and can be used as a diagnostic for the classification of models based on gravity theories even beyond general relativity [10, 11]. For a recent and well structured review about constraints for $\gamma$ in a wide range of models, see [11].

Another substantial advantage of Eq. (30) is that $\gamma$ happens to be quasi-constant for a wide range of models. [12] found a general expression for $\gamma(a)$ that applies to any model with a mixture of cold dark matter plus cosmological constant ($\Lambda$CDM) or quintessence (QCDM). For example, for a dark-energy equation of state parameterized by a slowly varying function $w(\Omega_\mathrm{m})$ in a spatially-flat universe, the growth index reduces to

$$\gamma = \frac{3(w - 1)}{6w - 5} \tag{31}$$

[13]. Thus, for any constant $w$, the growth index $\gamma$ is itself constant and reduces to $\gamma = 6/11$ for $\Lambda$CDM.

It is interesting in our context that we can derive $\gamma$ based on the reconstructed expansion function $E(a)$. As we show in Appendix B, an approximate, yet sufficiently accurate solution for $\gamma$ is

$$\gamma = \frac{\varepsilon + 3\omega}{2\varepsilon + 5\omega} \ . \tag{32}$$

For a $\Lambda$CDM model,

$$\frac{2aE'}{E} = \frac{2a}{E}\left(\frac{-3\Omega_\mathrm{m0}a^{-4}}{2E}\right) = -3\frac{\Omega_\mathrm{m0}}{E^2 a^3} = -3(1 - \omega) \ , \tag{33}$$

thus $\varepsilon = 3\omega$, and Eq. (32) reduces to $\gamma = 6/11$. With our reconstruction of the expansion function $E$, we can determine $\gamma$ and its uncertainty

$$\Delta\gamma = \left[\left(\frac{\partial\gamma}{\partial c_j}\right)^2 \Delta c_j^2\right]^{1/2} \tag{34}$$

for any choice of $\Omega_\mathrm{m0}$. The result for $\Omega_\mathrm{m0} = 0.3$ is shown for both data samples in Fig. 5.

The growth index follows the $\Lambda$CDM result very closely for $a \gtrsim 0.5$, but increases for smaller scale factors. Again, we abstain from drawing any conclusions here, but emphasise that our reconstruction method allows a direct determination of $\gamma$. It is likely that systematic errors in the data or any unaccounted covariance between the data points is responsible for the behaviour of $\gamma$ at $a \lesssim 0.5$.

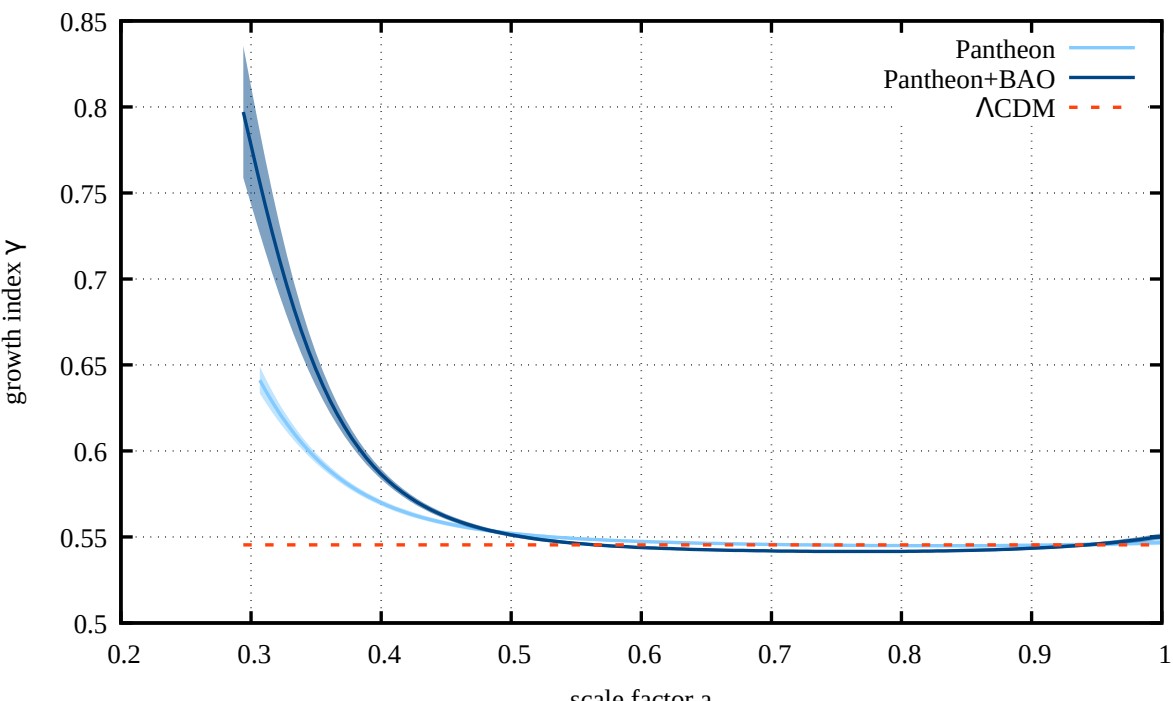

Figure 5: Growth index $\gamma$ derived from the expansion function $E$, reconstructed from the SN-sample and from the SN-BAO-sample, assuming $\Omega_{m0} = 0.3$.

## 4   Conclusions

We have shown here how the linear growth factor $D_+(a)$ of cosmic structures can be inferred from existing data with remarkably small uncertainty without reference to a specific cosmological model. Following up on, modifying and extending earlier studies, we have derived the cosmic expansion function $E(a)$ in a way independent of the cosmological model from the measurements of distance moduli to the type-Ia supernovae of the Pantheon sample (SN-sample), and from the Pantheon sample combined with a sample of BAO distance measurements compiled from the literature (SN-BAO-sample). All we need to assume is that underlying the cosmological model is a metric theory of gravity and that our universe satisfies the symmetry assumptions of spatial homogeneity and isotropy reasonably well. The uncertainty on this empirically determined expansion function already is remarkably small, and the results obtained from the SN-sample alone and from the SN-BAO-sample agree very well.

This expansion function is the main ingredient for the differential Eq. (25) describing cosmic structure growth in the linear limit. Only one further parameter is needed to solve this equation, viz. the present-day matter-density parameter $\Omega_{m0}$, because it enters into the initial conditions for solving Eq. (25). Assuming $\Omega_{m0}$, we can also solve for the growth index $\gamma$ defined in Eq. (30). This implies that, due to measurements of the distance moduli to the type-Ia supernovae in the SN- and SN-BAO-samples, the expansion function is accurately determined, and the linear growth factor $D_+$ as well as the growth index $\gamma$ are tightly constrained up to a single remaining parameter, i.e. the present-day matter density parameter $\Omega_{m0}$.

Comparing our results to the best-fitting expansion function of a spatially-flat, Friedmann-Lemaître model universe illustrated in Fig. 3, and the constraint on the growth index $\gamma$ shown in Fig. 5 demonstrate how our method can be used with future data to derive the possible time evolution of the dark energy

and the growth index directly from distance measurements.

In future work, we will extend the method presented here to further types of data. Our goal is to determine the two centrally important functions of cosmology, $E(a)$ and $D_+(a)$, with as few assumptions as possible and without reference to a specific cosmological model. Such applications of our results may be particularly interesting which so far require assuming cosmological parameters or models for a possible evolution of dark energy, e.g. cosmological weak gravitational lensing.

# Acknowledgements

It is a pleasure to thank Bettina Heinlein, Sven Meyer, Jiahan Shi, Lorenzo Speri, and Jenny Wagner for interesting and helpful discussions. This work was supported by the Deutsche Forschungsgemeinschaft (DFG) via the Transregional Collaborative Research Centre TRR 33 (MB, MM) and under Germany's Excellence Strategy EXC-2181/1 - 390900948 (the Heidelberg STRUCTURES Excellence Cluster).

# A    Chebyshev polynomials

The (unnormalised) Chebyshev polynomials of the first kind $\bar{T}_n(x)$ are defined on the interval $[-1, 1]$ by the recurrence relation

$$\bar{T}_{n+1}(x) = 2x\bar{T}_n(x) - \bar{T}_{n-1}(x) \tag{35}$$

with $\bar{T}_0(x) = 1$ and $\bar{T}_1(x) = x$. They can be written in the form

$$\bar{T}_n(\cos\theta) = \cos n\theta \tag{36}$$

and are orthogonal (but not orthornomal) with respect to the weight function $w(x) = (1 - x^2)^{-1/2}$,

$$
\begin{aligned}
\left\langle \bar{T}_n(x)\bar{T}_m(x) \right\rangle &= \int_{-1}^{1} \frac{\mathrm{d}x}{\sqrt{1-x^2}} \bar{T}_n(x)\bar{T}_m(x) = \int_0^{\pi} \mathrm{d}\theta \cos n\theta \cos m\theta \\
&= \begin{cases} 0 & n \neq m \\ \pi & n = m = 0 \\ \pi/2 & n = m \neq 0 \end{cases} .
\end{aligned}
\tag{37}
$$

The normalised Chebyshev polynomials are thus given by

$$T_n(x) := \begin{cases} (1/\pi)^{1/2} & (n = 0) \\ (2/\pi)^{1/2} \cos{(n \arccos x)} & (n > 0) \end{cases} . \tag{38}$$

Finally, the shifted Chebyshev polynomials are defined on the interval $[0, 1]$ in terms of the Chebyshev polynomials by

$$T_n^*(x) = T_n(2x - 1) . \tag{39}$$

They are orthonormal with respect to the weight function $w^*(x) = (x - x^2)^{-1/2}$.

## B  Derivation of the growth index

In terms of the logarithmic derivative

$$f := \frac{\mathrm{d}\ln D_+}{\mathrm{d}\ln a} \tag{40}$$

and using the parameters $\varepsilon$ and $\omega$ introduced in Eq. (27), the linear growth equation (24) reads

$$\frac{\mathrm{d}f}{\mathrm{d}\ln a} + \frac{1}{2}(1 + \varepsilon)f + f^2 = \frac{3}{2}(1 - \omega) . \tag{41}$$

We write

$$\frac{\mathrm{d}f}{\mathrm{d}\ln a} = f\frac{\mathrm{d}\ln\Omega_\mathrm{m}}{\mathrm{d}\ln a}\frac{\mathrm{d}\ln f}{\mathrm{d}\ln\Omega_\mathrm{m}} , \tag{42}$$

use Eq. (25) to find

$$\frac{\mathrm{d}\ln\Omega_\mathrm{m}}{\mathrm{d}\ln a} = -\varepsilon \tag{43}$$

and Eq. (30) to write

$$\frac{\mathrm{d}\ln f}{\mathrm{d}\ln\Omega_\mathrm{m}} = \gamma - \omega\frac{\mathrm{d}\gamma}{\mathrm{d}\ln\Omega_\mathrm{m}} , \tag{44}$$

approximating $\ln\Omega_\mathrm{m} = \ln(1 - \omega) \approx -\omega$ in the last step. Neglecting terms of order $\varepsilon\omega$, we have

$$\frac{\mathrm{d}f}{\mathrm{d}\ln a} = -\varepsilon\gamma f . \tag{45}$$

Inserting this result into Eq. (41), dividing by $f$ and approximating

$$f = \Omega_\mathrm{m}^\gamma = (1 - \omega)^\gamma \approx 1 - \gamma\omega , \tag{46}$$

we arrive at

$$-\varepsilon\gamma + \frac{1}{2}(1 + \varepsilon) + 1 - \gamma\omega = \frac{3}{2}\left[1 + (\gamma - 1)\omega\right] \tag{47}$$

to linear order in $\varepsilon$ and $\omega$. Solving for $\gamma$ finally gives the result

$$\gamma = \frac{\varepsilon + 3\omega}{2\varepsilon + 5\omega} \tag{48}$$

quoted in Eq. (32).

## C  BAO sample

The sample of BAO measurements collected from the literature is listed in Tab. 2.

Table 2: BAO data

| $n$ | $z$ | $D_A/r_d$ | $\Delta(D_A/r_d)$ | Description | Reference |
|---|---|---|---|---|---|
| 1 | 0.240 | 5.3637 | 0.4673 | autocorrelation function of CMASS galaxies in BOSS DR12 | [20] |
| 2 | 0.240 | 5.5939 | 0.3048 | redshift-space distortion moments of LOWZ and CMASS galaxy samples in BOSS DR12 | [21] |
| 3 | 0.310 | 6.2900 | 0.1400 | tomographic configuration-space analysis of galaxy autocorrelations in BOSS DR12 | [22] |
| 4 | 0.310 | 6.2948 | 0.1963 | tomographic analysis of galaxy clustering in BOSS DR12 | [23] |
| 5 | 0.310 | 6.3045 | 0.2734 | tomographic analysis of redshift-space distortion moments in BOSS DR12 galaxies | [24] |
| 6 | 0.320 | 6.6978 | 0.2099 | autocorrelation function of CMASS galaxies in BOSS DR12 | [20] |
| 7 | 0.320 | 6.4743 | 0.1896 | redshift-space distortion moments of LOWZ and CMASS galaxy samples in BOSS DR12 | [21] |
| 8 | 0.320 | 6.6689 | 0.3943 | autocorrelation function of CMASS and LOWZ galaxies in BOSS DR12, z = 0.3-0.5 | [25] |
| 9 | 0.320 | 6.6600 | 0.1600 | analysis of redshift-space distortion moments in BOSS DR14 quasars | [26] |
| 10 | 0.360 | 7.0900 | 0.1600 | tomographic configuration-space analysis of galaxy autocorrelations in BOSS DR12 | [22] |
| 11 | 0.360 | 6.9379 | 0.2572 | tomographic analysis of galaxy clustering in BOSS DR12 | [23] |
| 12 | 0.360 | 7.0870 | 0.2390 | tomographic analysis of redshift-space distortion moments in BOSS DR12 galaxies | [24] |
| 13 | 0.370 | 7.3818 | 0.3318 | autocorrelation function of CMASS galaxies in BOSS DR12 | [20] |
| 14 | 0.370 | 6.7249 | 0.4402 | redshift-space distortion moments of LOWZ and CMASS galaxy samples in BOSS DR12 | [21] |
| 15 | 0.380 | 7.4435 | 0.2730 | galaxy clustering in BOSS DR12, combined with various priors | [27] |
| 16 | 0.380 | 7.3894 | 0.1218 | power spectrum of galaxy distribution in BOSS DR12 | [28] |
| 17 | 0.380 | 7.3894 | 0.1116 | galaxy clustering in BOSS DR12, systematic-error analysis | [29] |
| 18 | 0.400 | 7.7000 | 0.1600 | tomographic configuration-space analysis of galaxy autocorrelations in BOSS DR12 | [22] |
| 19 | 0.400 | 7.5335 | 0.2166 | tomographic analysis of galaxy clustering in BOSS DR12 | [23] |
| 20 | 0.400 | 7.6576 | 0.2407 | tomographic analysis of redshift-space distortion moments in BOSS DR12 galaxies | [24] |
| 21 | 0.440 | 8.2000 | 0.1300 | tomographic configuration-space analysis of galaxy autocorrelations in BOSS DR12 | [22] |
| 22 | 0.440 | 8.0547 | 0.1760 | tomographic analysis of galaxy clustering in BOSS DR12 | [23] |
| 23 | 0.440 | 8.0464 | 0.1601 | tomographic analysis of redshift-space distortion moments in BOSS DR12 galaxies | [24] |
| 24 | 0.450 | 8.2881 | 0.2954 | angular galaxy clustering in SDSS DR10 | [30] |
| 25 | 0.470 | 7.7682 | 0.3869 | angular galaxy clustering in SDSS DR10 | [30] |

Table 2: BAO data (continued)

| $n$ | $z$ | $D_A/r_d$ | $\Delta(D_A/r_d)$ | Description | Reference |
|---|---|---|---|---|---|
| 26 | 0.480 | 8.6400 | 0.1100 | tomographic configuration-space analysis of galaxy autocorrelations in BOSS DR12 | [22] |
| 27 | 0.480 | 8.6977 | 0.1895 | tomographic analysis of galaxy clustering in BOSS DR12 | [23] |
| 28 | 0.480 | 8.6059 | 0.1812 | tomographic analysis of redshift-space distortion moments in BOSS DR12 galaxies | [24] |
| 29 | 0.490 | 7.7100 | 0.3245 | angular galaxy clustering in SDSS DR10 | [30] |
| 30 | 0.490 | 8.7092 | 0.2641 | autocorrelation function of CMASS galaxies in BOSS DR12 | [20] |
| 31 | 0.490 | 8.7227 | 0.2099 | redshift-space distortion moments of LOWZ and CMASS galaxy samples in BOSS DR12 | [21] |
| 32 | 0.510 | 7.8926 | 0.2789 | angular galaxy clustering in SDSS DR10 | [30] |
| 33 | 0.510 | 8.8510 | 0.1264 | galaxy clustering in BOSS DR12, systematic-error analysis | [29] |
| 34 | 0.520 | 8.9000 | 0.1200 | tomographic configuration-space analysis of galaxy autocorrelations in BOSS DR12 | [22] |
| 35 | 0.520 | 9.0565 | 0.2031 | tomographic analysis of galaxy clustering in BOSS DR12 | [23] |
| 36 | 0.520 | 9.0465 | 0.1984 | tomographic analysis of redshift-space distortion moments in BOSS DR12 galaxies | [24] |
| 37 | 0.530 | 8.7336 | 0.6107 | angular galaxy clustering in SDSS DR10 | [30] |
| 38 | 0.550 | 8.7021 | 0.5119 | angular galaxy clustering in SDSS DR10 | [30] |
| 39 | 0.560 | 9.1600 | 0.1400 | tomographic configuration-space analysis of galaxy autocorrelations in BOSS DR12 | [22] |
| 40 | 0.560 | 9.3813 | 0.2031 | tomographic analysis of galaxy clustering in BOSS DR12 | [23] |
| 41 | 0.560 | 9.3778 | 0.2077 | tomographic analysis of redshift-space distortion moments in BOSS DR12 galaxies | [24] |
| 42 | 0.570 | 9.5241 | 0.1428 | autocorrelation function of CMASS and LOWZ galaxies in BOSS DR12, z = 0.3-0.5 | [25] |
| 43 | 0.570 | 9.4200 | 0.1300 | analysis of redshift-space distortion moments in BOSS DR14 quasars | [26] |
| 44 | 0.590 | 9.5896 | 0.1693 | autocorrelation function of CMASS galaxies in BOSS DR12 | [20] |
| 45 | 0.590 | 9.6235 | 0.1558 | redshift-space distortion moments of LOWZ and CMASS galaxy samples in BOSS DR12 | [21] |
| 46 | 0.590 | 9.4500 | 0.1700 | tomographic configuration-space analysis of galaxy autocorrelations in BOSS DR12 | [22] |
| 47 | 0.590 | 9.5167 | 0.2301 | tomographic analysis of galaxy clustering in BOSS DR12 | [23] |
| 48 | 0.590 | 9.6347 | 0.2279 | tomographic analysis of redshift-space distortion moments in BOSS DR12 galaxies | [24] |
| 49 | 0.610 | 9.6292 | 0.1593 | galaxy clustering in BOSS DR12, systematic-error analysis | [29] |
| 50 | 0.640 | 9.9011 | 0.2844 | autocorrelation function of CMASS galaxies in BOSS DR12 | [20] |

Table 2: BAO data (continued)

| $n$ | $z$ | $D_A/r_d$ | $\Delta(D_A/r_d)$ | Description | Reference |
|---|---|---|---|---|---|
| 51 | 0.640 | 9.7792 | 0.2777 | redshift-space distortion moments of LOWZ and CMASS galaxy samples in BOSS DR12 | [21] |
| 52 | 0.640 | 9.6200 | 0.2200 | tomographic configuration-space analysis of galaxy autocorrelations in BOSS DR12 | [22] |
| 53 | 0.640 | 9.5573 | 0.2775 | tomographic analysis of galaxy clustering in BOSS DR12 | [23] |
| 54 | 0.640 | 9.8065 | 0.3849 | tomographic analysis of redshift-space distortion moments in BOSS DR12 galaxies | [24] |
| 55 | 0.800 | 10.3720 | 0.9699 | Fourier-space measurement of clustering of eBOSS DR14 quasars | [31] |
| 56 | 0.800 | 10.8119 | 1.1428 | clustering of 147000 eBOSS DR14 quasars | [32] |
| 57 | 0.978 | 10.7334 | 1.9281 | tomographic analysis of quasar clustering in eBOSS DR14 | [33] |
| 58 | 1.000 | 12.0449 | 0.9880 | Fourier-space measurement of clustering of eBOSS DR14 quasars | [31] |
| 59 | 1.000 | 11.5205 | 1.0319 | clustering of 147000 eBOSS DR14 quasars | [32] |
| 60 | 1.230 | 11.9710 | 1.0805 | tomographic analysis of quasar clustering in eBOSS DR14 | [33] |
| 61 | 1.500 | 12.0693 | 0.7443 | Fourier-space measurement of clustering of eBOSS DR14 quasars | [31] |
| 62 | 1.500 | 12.1559 | 0.7362 | clustering of 147000 eBOSS DR14 quasars | [32] |
| 63 | 1.520 | 12.5186 | 0.7443 | combination of power spectrum and bispectrum of BOSS DR12 galaxies | [34] |
| 64 | 1.520 | 12.5186 | 0.6767 | clustering of 148659 quasars from eBOSS DR14 survey | [35] |
| 65 | 1.526 | 11.9689 | 0.6536 | tomographic analysis of quasar clustering in eBOSS DR14 | [33] |
| 66 | 1.944 | 12.2343 | 0.9911 | tomographic analysis of quasar clustering in eBOSS DR14 | [33] |
| 67 | 2.000 | 12.3585 | 0.5391 | Fourier-space measurement of clustering of eBOSS DR14 quasars | [31] |
| 68 | 2.000 | 12.0111 | 0.5616 | clustering of 147000 eBOSS DR14 quasars | [32] |
| 69 | 2.200 | 12.1697 | 0.4969 | Fourier-space measurement of clustering of eBOSS DR14 quasars | [31] |
| 70 | 2.200 | 11.8546 | 0.5392 | clustering of 147000 eBOSS DR14 quasars | [32] |
| 71 | 2.225 | 10.0425 | 1.7588 | autocorrelation function of BOSS DR12 quasars | [36] |
| 72 | 2.330 | 11.3423 | 0.6396 | Lya forest in 157783 BOSS DR12 quasars | [37] |
| 73 | 2.340 | 11.2754 | 0.6513 | Lya forest in 137562 BOSS DR11 quasars | [38] |
| 74 | 2.360 | 10.8000 | 0.4000 | Lya forest in 137562 BOSS DR11 quasars | [38] |
| 75 | 2.400 | 10.5000 | 1.2513 | cross-correlation between 234367 quasars and 168889 forests in BOSS | [39] |

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
