# Peer review of "Model-Independent Determination of the Cosmic Growth Factor"

_SciPost Astronomy_

## Round 1 · Referee Report · Anonymous (Referee 1) · 2020-3-26

Report

In the paper "Model-Independent Determination of the Cosmic Growth Factor" (arxiv: 1912.04560) the authors reconstructed expansion function and the linear growth function using Pantheon and SN-BAO samples. Authors argue that the results can be parameterized by a single number (present-day matter-density parameter).
Before the paper can be accepted for a publication I suggest authors to address the following points:

1. Pantheon sample has been criticized by M. Rameez et al in arxiv:1911.06456. I recommend the authors to address and/or account for points presented in this paper

2. Eq. 18 looks for me as a simplified form of Eq. 1 in which radiation and curvature terms are neglected. If this is the case it is trivial that the expansion function and any derived quantities are characterized by a matter density parameter only; I recommend authors to clarify this point.

I am not sure that in this case the uncertainty on $\Omega_{m0}$ has physical meaning. It is formally correct, but seems to rely on neglecting of radiation/curvature densities which are by an order of magnitude comparable to the reported uncertainty.

3. I suggest authors to consider replace Fig.1-2 with log-log scale versions to make uncertainty region a bit more clear. Please also refer in Fig.3 to eq. 22 which provides the definition of "dark energy evolution". Please change also y-label in this Fig. accordingly.

---

## Editorial Decision

resubmitted